# The Effect of Electrical Muscle Stimulation on Muscle Mass and Balance in Older Adults with Dementia

**DOI:** 10.3390/brainsci11030339

**Published:** 2021-03-07

**Authors:** Yuichi Nishikawa, Tetsuya Takahashi, Shuhei Kawade, Noriaki Maeda, Hirofumi Maruyama, Allison Hyngstrom

**Affiliations:** 1Faculty of Frontier Engineering, Institute of Science & Engineering, Kanazawa University, Kanazawa 920-1192, Japan; 2Department of Rehabilitation, Faculty of Rehabilitation, Hiroshima International University, Hiroshima 739-2695, Japan; tetakaha@me.com; 3MTG Co., Ltd., Nagoya 453-0041, Japan; shuhei.kawade@mtg.gr.jp; 4Division of Sports Rehabilitation, Graduate School of Biomedical and Health Sciences, Hiroshima University, Hiroshima 734-8551, Japan; norimmi@hiroshima-u.ac.jp; 5Department of Clinical Neuroscience and Therapeutics, Graduate School of Biomedical and Health Sciences, Hiroshima University, Hiroshima 734-8551, Japan; hmaru@hiroshima-u.ac.jp; 6Department of Physical Therapy, Marquette University, Milwaukee, WI 53201, USA; allison.hyngstrom@marquette.edu

**Keywords:** elderly people, electrical muscle stimulation, muscle mass, dementia

## Abstract

Background: Electrical muscle stimulation (EMS) is effective for increasing physical function. However, there is no evidence regarding the effects of EMS on muscle mass and physical function in older adults with dementia. The aim of the present study was to quantify the effects of EMS on muscle mass and balance in older adults with dementia. Methods: A total of 32 participants were randomly assigned to an intervention group (*n* = 16, age = 89.4 ± 4.8 years) and a control group (*n* = 16, age = 88.1 ± 5.2 years). Participants in the intervention group underwent a general rehabilitation program (20 min for three days/week) and an EMS intervention (23 min for three days/week) for 23 weeks. Participants in the control group underwent general rehabilitation only. The efficacy of EMS was evaluated by lower limb muscle mass, the Berg Balance Scale (BBS), and the functional independence measure (FIM). Results: Muscle mass was significantly increased in the intervention group after 12 weeks (*p* = 0.008), but average muscle mass in the control group did not change (*p* = 0.18). Participants in the control group showed a significant decrease in BBS after 12 weeks (*p* = 0.007), unlike those in the intervention group. Furthermore, there was a strong correlation between the mini-mental state examination (MMSE) results and the change in muscle mass, the BBS, and the FIM in the control group (*p* < 0.05). Conclusions: These findings suggest that EMS is a useful intervention for increasing muscle mass and maintaining balance function in older adults with dementia.

## 1. Introduction

The Japanese population is aging rapidly. The population of elderly individuals aged 65 years or older is expected to reach 39.35 million by 2042. Furthermore, it is estimated that the aging rate (defined as the share of the population aged 65 or over) in Japan will reach 38.4% in 2065 [1]. Aging causes a progressive loss of muscle mass, which results in decreased muscle strength and physical performance [2]. A previous study reported that age has a negative impact on strength and physical performance in independent, healthy elderly women, and that physical activity test (e.g., Berg Balance Scale (BBS) and Timed Up and Go test) outcomes are positively influenced by muscle mass and strength [3]. The number of individuals who require long-term care (e.g., nursing care) is increasing rapidly among the elderly population, especially those aged75 years or older [1]. Therefore, in Japan, there is an urgent need to improve the physical function of older adults in order to decrease the number of individuals requiring long-term care. Previous studies have shown that rehabilitation is an effective method of improving physical function and quality of life in older adults [4,5]. However, there have been few reports on those with dementia and/or aged 80 years or older. Many older adults in nursing homes are over 80 years of age, and many have dementia and/or are bedridden, which might limit their ability to participate in conventional rehabilitation. Effective rehabilitation interventions for this population have not been established.

Electrical muscle stimulation (EMS) is an attractive alternative method of inducing muscle contraction, thereby serving as a surrogate for habitual physical activity during periods of muscle disuse due to illness or injury [6]. Hortobágyi and Maffiuletti et al. reported that EMS modifies the excitability of specific neural pathways and increases maximal voluntary force without apparent muscle hypertrophy [7]. Research has shown that EMS may improve muscle mass [8] and reverse muscle atrophy caused by inactivity without voluntary effort [9]. A previous study also reported that EMS is effective in increasing muscle strength, muscle thickness, and activities of daily living (ADLs) in older adults [10]. However, there is no evidence regarding the effects of EMS on muscle mass or motor function in older adults with dementia. Since EMS can directly stimulate muscle fibers, one advantage of its use is that it can be used in individuals who have difficulty with voluntary muscle activation or have cognitive impairment, which limits their ability to participate in conventional rehabilitation. We have previously reported that EMS interventions on the thigh in elderly people did not improve balance performance (single leg standing) [10]. In addition to the thigh muscles, the plantar intrinsic muscles are also important for balance function [11,12]. These findings indicate the importance of interventions on the plantar intrinsic muscles to promote balance function. Therefore, we focused on EMS of the foot sole to promote muscle contraction in the entire lower limb including the thigh and lower leg. An EMS device (SIXPAD Foot fit, MTG Co., Ltd., Nagoya, Japan) was used to apply EMS to the foot sole to stimulate the contraction of muscles in the entire lower limb, mainly the plantar intrinsic muscles, tibialis anterior muscle, and triceps surae muscle. We hypothesized that this EMS intervention would be more effective than thigh stimulation in improving performance in the balance task.

The purpose of the present study was to examine the efficacy of a combined EMS intervention on muscle mass and balance performance improvement in older adults with dementia. We hypothesized that compared with general rehabilitation, a combined EMS intervention would better elicit increased muscle mass and balance function.

## 2. Materials and Methods

### 2.1. Trial Design and Setting

This was a single-center, single-blind (examiner), parallel-group study with a 1:1 allocation ratio. The trial was prospectively registered with the Japan Primary Registries Network (UMIN Clinical Trials Registry; No. UMIN000026253).

### 2.2. Participants

A total of 32 participants were enrolled in this study and randomly assigned to the intervention group (*n* = 16, age = 89.4 ± 4.8 years old, height = 145.2 ± 6.9 cm, weight = 45.9 ± 5.5 kg, and Mini-Mental State Examination (MMSE) score = 2–20 [min–max]) or the control group (*n* = 16, age = 88.1 ± 5.2 years old, height = 147.3 ± 5.8 cm, weight = 47.0 ± 4.9 kg, and MMSE score = 2–20). The inclusion criteria were that participants be older adults (80 years or older) who had dementia (MMSE score < 20) and were undergoing rehabilitation in a nursing home. The exclusion criteria were cardiovascular disease, consciousness disorders, and diabetes mellitus. All procedures were performed in accordance with the Declaration of Helsinki and approved by Hiroshima University’s Committee on Ethics in Research (approval no. C-151). All participants and their families signed informed-consent forms and consented to the publication of this work.

### 2.3. Randomization and Allocation Procedure

Computer-generated random numbers were used for simple randomization of the participants. Only one independent researcher, who was not involved in recruitment, assessments, or treatment, performed the randomization of participants after recruitment and concealed the randomization from the examiner.

### 2.4. Experimental Design

Participants in the intervention group underwent bilateral EMS of the lower limb muscles in a sitting position for 23 min once per day, three days per week, for 12 weeks using an EMS device (SIXPAD Foot fit, MTG Co., Ltd., Nagoya, Japan). EMS was applied to bare feet (Appendix A). Muscles were stimulated at an intensity of 4.85 mA (frequency = 20 Hz, pulse shape = square wave, pulse duration = 100 µs, and pulse period = 50 ms) [10]. Participants were instructed not to actively contract their muscles during the stimulation protocol. Participants in the intervention group also underwent a general rehabilitation program that was conducted by physical and/or occupational therapists and involved transfer, balance, and gait training. Transfer training was based on moving from a sitting to a standing position. Balance training included a single-leg standing posture and standing on an unbalanced surface. Gait training included walking on uneven ground and floors using a cane and/or parallel bar. The EMS intervention and general rehabilitation program were performed on the same days. Participants in the control group only underwent a general rehabilitation program. To quantify the effects of EMS, the following variables were measured at baseline and 12 weeks by a blinded examiner: muscle mass, BBS score, MMSE score, and functional independence measure (FIM) score. We calculated the percent change in the BBS score, MMSE score, and FIM score from baseline at 12 weeks. Baseline assessments were conducted one week prior to the start of the intervention, and post-intervention assessments were conducted within one week after the end of the intervention.

### 2.5. Measurements of Muscle Mass

Measurements of lower-limb (bilateral-thigh and lower-leg) muscle mass were performed using direct segmental multifrequency bioelectrical impedance analysis (InBody S10, InBody Japan, Tokyo, Japan). This method of estimating skeletal muscle mass has been validated with dual-energy X-ray absorptiometry [13]. Electrodes were placed bilaterally on the thumbs, middle fingers, and ankles. Participants were asked to lie down in a supine position and instructed to lie as still as possible during the measurements. Measurements were obtained 5 hours before meals.

### 2.6. Statistical Analysis

Statistical analyses were performed using SPSS version 25.0 (SPSS, Inc., Chicago, IL, USA). The mean, standard deviation (SD), and 95% confidence interval (CI) of the values were calculated for each variable. Before the analysis, the normality of the data distribution was confirmed using the Shapiro–Wilk test. Age, height, body mass, MMSE score, FIM score, BBS score, and muscle mass were compared between the intervention and control groups using unpaired *t*-tests. Muscle mass, the BBS score, and the FIM score were analyzed by mixed analysis of variance (ANOVA) (two-way factor; group (intervention and control) vs. period (pre and post)). If a significant interaction effect was observed, multiple comparisons with Bonferroni adjustments were performed. Two-tailed *p* values < 0.05 were considered statistically significant. The effect size was calculated using the difference between two means divided by the pooled SD. The interpretation of the effect size was performed on the basis of Cohen’s recommendation, where *d* = 0.20 indicates a small effect size, *d* = 0.50 indicates a medium effect size, and *d* = 0.80 indicates a large effect size [14]. Pearson’s correlation coefficients were computed to assess bivariate correlations among the MMSE score, FIM score, ∆muscle mass, and ∆BBS score. The strength of the correlation coefficients was qualitatively interpreted according to the following thresholds: 0.2 ≤ *r* ≤ 0.4, small; 0.4 ≤ *r* ≤ 0.7, moderate; 0.7 ≤ *r* ≤ 0.9, strong; and 0.9 ≤ *r* ≤ 1.0, very strong [15]. We used a Bonferroni correction for Pearson’s correlation coefficients (critical alpha < 0.05/2 = 0.025).

## 3. Results

This trial was conducted between February 2017 and December 2019 (end of participant follow-up). The general characteristics of the subjects are shown in Table 1. There were no significant differences between the groups in terms of anthropometric parameters. A total of 21 participants completed the intervention and evaluations (the intervention group, *n* = 11; death, *n* = 2; and drop out, *n* = 3; and the control group, *n* = 10; death, *n* = 2; and drop out, *n* = 4, Figure 1). There were no significant differences in these parameters between the subjects who dropped out and those who completed the intervention (Appendix A). The cause of death was aspiration pneumonia, and the cause of dropout was transfer to other hospitals.

The MMSE scores did not change pre- to postintervention in either group. Muscle mass and BBS scores showed significant interactions between group and period (*F* = 10.287, *p* = 0.004 and *F* = 5.834, *p* = 0.0260, respectively). The intervention group showed significantly higher muscle mass postintervention than preintervention (7.57 ± 1.60 kg to 8.07 ± 1.93 kg, 95% CI = 0.18 to 0.69, *p* = 0.008, and effect size = 0.51), unlike the control group (8.04 ± 1.30 kg to 7.63 ± 1.38 kg, 95% CI = −0.72 to 0.07, *p* = 0.176, and effect size = 0.17) (Figure 2A). Five people in the control group had decreased BBS and FIM scores, but none did in the intervention group. The control group showed significantly decreased BBS scores postintervention compared with preintervention (21.81 ± 11.48 to 19.64 ± 13.38, *p* = 0.0071, 95% CI = −1.26 to −0.02, and effect size = 0.59), unlike the intervention group (22.19 ± 14.10 to 23.00 ± 15.77, *p* = 0.341, 95% CI = −0.29 to 0.11, and effect size = 0.30) (Figure 2B). The FIM score did not show a significant interaction between group and period (F = 0.528, *p* = 0.476) or by group (*p* = 0.729, 95% CI = −4.62 to 6.52) or period (*p* = 0.702, 95% CI = −1.89 to 2.76) (Figure 2C).

In the control group, there were strong correlations between the MMSE score and *Δ*muscle mass, ΔFIM sore, and ΔBBS score (*r* = 0.6945 and *p* = 0.0177, *r* = 0.7140 and *p* = 0.0136, and *r* = 0.7077 and *p* = 0.0148, respectively) (Figure 3). In the intervention group, there were no correlations between MMSE score and Δmuscle mass or ΔFIM score (r = −0.1696 and *p* = 0.6181, *r* = −0.01187 and *p* = 0.9724) (Figure 3A,B, respectively). Finally, the control group showed significant correlations between the baseline FIM score and ΔBBS score and Δmuscle mass (*r* = 0.7396, *p* = 0.00093 and *r* = 0.7643, *p* = 0.0062, respectively) (Figure 4); this finding was not observed in the intervention group (*r* = −0.1324, *p* = 0.6980) (Figure 4B).

## 4. Discussion

In this study, we demonstrated the effects of a combined EMS intervention on muscle mass, the BBS score, and the FIM score in older adults with dementia. The primary findings of the present study were that a combined EMS intervention on the lower limbs increased muscle mass and prevented balance decline or maintained current balance function in adults aged over 80 years with dementia. On the other hand, in the control group, muscle mass did not change, and decreases in BBS and FIM scores were observed. There were strong correlations between (1) the MMSE score and Δmuscle mass, ΔFIM score, and ΔBBS score, and (2) the FIM score and Δmuscle mass and ΔBBS score in the control group.

The present study revealed a significant increase in muscle mass and maintenance of balance function after the combined EMS intervention. Notably, the intervention group did not show a decline in balance function, even in subjects with low MMSE scores (<10) and FIM scores (<50). It is difficult to maintain muscle mass and balance function in individuals whose cognitive and/or physical function have deteriorated. Previous studies have reported that rehabilitation leads to improved quality of life and motivation to exercise in older adults with dementia [16,17]. However, cognitive function is an important factor in the rehabilitation process, and it influences the efficacy of physiotherapy [18]. A previous study reported that moderate- to high-intensity training programs did not result in improvement in ADLs or cognitive function in order adults with dementia [19]. It is particularly difficult to improve function in bedridden older adults because effective training cannot be performed. Despite undergoing rehabilitation, individuals in the control group showed deceased balance performance and no increase in muscle mass. However, our study included bedridden subjects in each group, and the increase in muscle mass and maintenance of balance scores in these subjects in the intervention group were very meaningful results.

It is widely known that EMS interventions can improve muscle performance [20,21,22,23,24]. Many previous studies have also reported improvements in muscle strength and thickness after EMS interventions [10,25,26,27]. Furthermore, in our previous study, EMS interventions enhanced muscle strength and muscle thickness in older adults (age, 75.6 ± 3.7 years) [10]. These previous findings are in accordance with the results of the present study, in which individuals in the intervention group showed significantly increased muscle mass and maintenance of BBS scores compared with those in the control group. These findings suggest that EMS is an effective treatment for older adults with dementia, including bedridden individuals. To the best of our knowledge, the present study is the first to report an effective method for improving physical performance in older adults with dementia (over 85 years old). However, our findings did not show improved FIM scores (ADL levels) in either group. This study included older adults who had dementia and were bedridden. Therefore, the study period may have been too short to show changes in physical function. Future studies should consider a combination of different interventions, rather than general rehabilitation, for a longer period (e.g., high-intensity muscle strength training and virtual reality training).

Our study had several limitations. First, it was performed at a single center with a small sample. As mortality increases with age, multicenter interventions will be necessary in the future. Second, the present study performed EMS on lower limb muscles only. A previous study that applied EMS to lower limb muscles, including those of the trunk, showed that by applying stimulation interventions that include more proximal muscle, it is possible to influence standing balance control. In future research, it will be necessary to implement EMS over a wider area. Finally, this study did not assess muscle strength or walking performance, both of which are difficult to assess in patients with dementia who are bedridden. Future research needs to include healthy older adults with no dementia.

## 5. Conclusions

An EMS intervention increased muscle mass and promoted maintenance of balance performance in older adults with dementia. Our results support our hypothesis that EMS intervention on the plantar intrinsic muscles can produce favorable results in terms of balance function. Furthermore, this study demonstrates that EMS interventions can be performed in not only healthy individuals but also those with dementia without adverse events. These findings suggest that EMS is a useful technique for maintaining and improving function in older adults with dementia. Further studies will need to examine the area over which EMS is applied, the frequency of application, and the duration of EMS for effective intervention.

## Figures and Tables

**Figure 1 brainsci-11-00339-f001:**
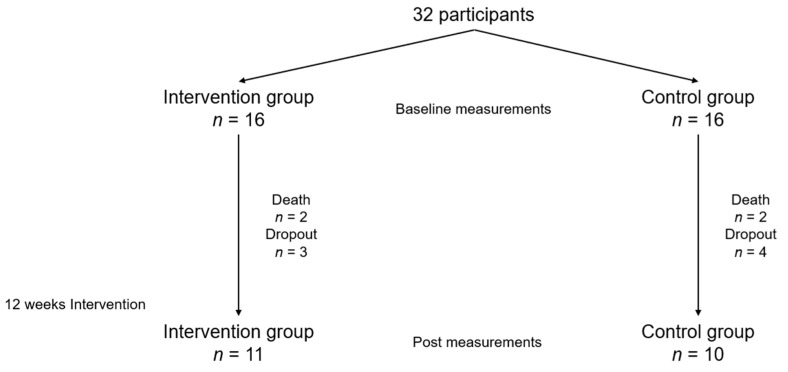
Flow chart for this study.

**Figure 2 brainsci-11-00339-f002:**
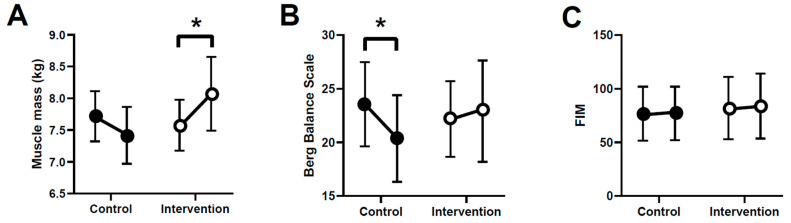
Comparison of muscle mass (**A**), Berg Balance Scale (BBS) score (**B**), and FIM score (**C**) pre- and postintervention in the two groups. * *p* < 0.05. Data shown means ± standard error of the mean.

**Figure 3 brainsci-11-00339-f003:**
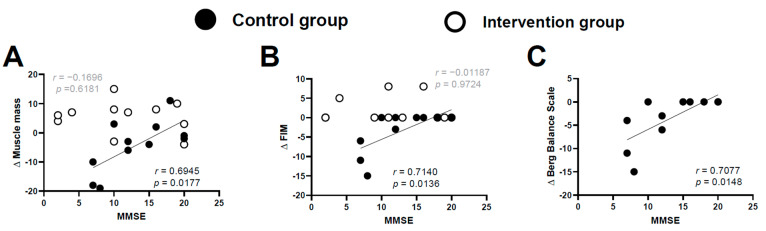
Correlations between the MMSE score and Δmuscle mass (**A**), ΔFIM (**B**), and ΔBBS score (**C**). Strong correlations were observed between the MMSE score and Δmuscle mass, ΔFIM score, and ΔBBS score in the control group (Bonferroni correction; critical alpha < 0.05/2 = 0.025). Data from the intervention group are not shown in Figure C because none of the individuals in this group had a decreased BBS score.

**Figure 4 brainsci-11-00339-f004:**
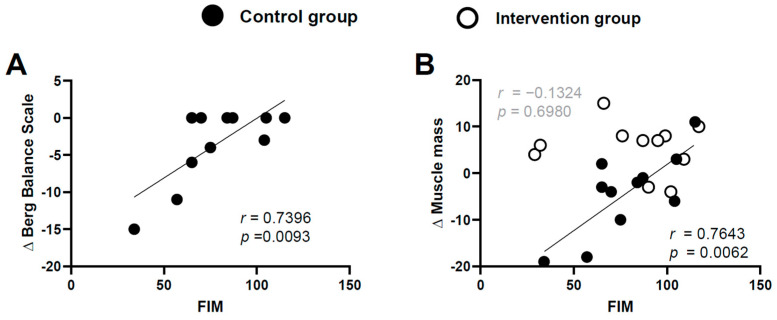
Correlations between the FIM score and ΔBBS (**A**) and Δmuscle mass (**B**) in the control group and the intervention group. Strong correlations were observed between the FIM score and Δmuscle mass and ΔBBS in the control group; this finding was not observed for the intervention group (Bonferroni correction; critical alpha < 0.05/2 = 0.025). Data from the intervention group are not shown in Figure A because none of the individuals in this group had a decreased BBS score.

**Table 1 brainsci-11-00339-t001:** Characteristics of subjects.

Variables	Intervention Group	Control Group	*p* Value
Age, years	89.4 ± 4.8	88.1 ± 5.2	*p* = 0.4949
Height, cm	145.2 ± 6.9	147.3 ± 5.8	*p* = 0.3577
Body mass, kg	45.9 ± 5.5	46.9 ± 4.9	*p* = 0.5803
MMSE	12.4 ± 7.2 (5–20)	13.4 ± 5.1 (6–20)	*p* = 0.6381
FIM	83.8 ± 29.0 (29–117)	76.8 ± 25.1 (25–115)	*p* = 0.5884
Berg Balance Score	22.5 ± 14.1	21.8 ± 11.5	*p* = 0.8892
Lower limb muscle mass, kg	7.6 ± 1.6	8.0 ± 1.3	*p* = 0.3698

Data are presented as the mean ± SD, (min–max). Mini-mental state examination, MMSE; functional independence measure, FIM.

## Data Availability

Data are available via request from the corresponding author (Y.N.).

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
