# Peer review of "The Effect of Electrical Muscle Stimulation on Muscle Mass and Balance in Older Adults with Dementia"

_brainsci, 2021, doi:10.3390/brainsci11030339_

Round 1

Reviewer 1 Report

Summary:

This is a very interesting topic and the manuscript is very clearly written and laid out. I commend the authors for the thought, time, and effort that went into this study, it was a pleasure to read!  My only general comment is that the manuscript could benefit from added background information (in the introduction and conclusion) to help set up and support the study hypotheses. Some minor clarifications and suggestion are also listed below.

General:

Introduction: The background information regarding EMS training is limited to 4 sentences and 3 references. This is doing a disservice to the field of research that currently exists on this topic. I would like to see the introduction greatly expanded upon with regards to EMS training and mechanisms (consider including the work of Hortobagyi & Maffiuletti, 2011, EJAP). Additionally, the authors don’t give any reason as to why EMS was applied to the foot sole (rather than the lower or upper legs, for example). This would be a good opportunity to provide rationale for that decision.

The conclusion is lacking in depth. The fact that the control group but not the intervention group demonstrated significant correlations should be investigated further. The conclusion would also benefit from background research on the mechanisms of EMS training and why the hypotheses were made; as well as information on how other training forms have impacted muscle mass, BBS, and FIM in older adults.

Specific:

Experimental Design (line 95): please clarify here if participants were already participating in an exercise intervention (such as the transfer, balance, and gait training) prior to being enrolled in the study, or if all interventions were novel upon recruitment.  It may be helpful to describe participants baseline ‘exercise’ status prior to recruitment, if possible (e.g., participation in nursing home rehabilitation or activities).

Experimental Design (line 96): please clarify here if the EMS and general rehab training were performed in tandem, on the same days but with separation, on different days, etc.

Experimental Design (line 103): please clarify if the measurements (baseline and 12-weeks) were conducted on days when participants had completed training (EMS or general rehab) or not.

Muscle mass (line 107): please clarify the segmental muscle mass that was calculated (i.e., right and left legs from hip to toe; from knee to toe; whole body?).

Statistics: Please clarify if the MMSE score (taken a baseline and 12-weeks) was examined using an ANOVA; and if not, why?

Figure 2: the small figures, scale, and small adaptations pre-post make it very difficult to see any changes present between groups. Please consider changing the types of graphs (e.g., line graphs from pre-post with a solid vs dotted line for control vs intervention, or bar graph with control vs intervention on the x-axis and the pre-post bars side-by-side for better comparison).

Figures 3 & 4 and respective text: it is unclear why the correlation between BBS and MMSE in the intervention group (fig 3c) and BBS and FIM in the intervention group (fig 4a) are not included in the figures.

Line 184: this is very oddly worded “… and did not decrease balance function in older adults…”. This implies that your hypothesis was that EMS training would decrease balance function. It was not a function of the control intervention (i.e., general rehab) that decreased balance function, but likely a function of the added EMS training which prevented balance decline. This should be reworded to not that EMS training may prevent balance decline or maintain current balance function.

Line 215: This sentence requires greater clarity – what do you mean by ‘did not show improved physical function’? Does this refer specifically to the FIM measure. And to further clarify, please specify by intervention group that you mean the EMS training group.

Author Response

Response to reviewer

We thank the reviewers for their thorough review of our manuscript. We have substantially revised our manuscript according to the suggestions of the reviewer. Point-by-point responses to each of the reviewers’ comments are provided below.

General:

Introduction: The background information regarding EMS training is limited to 4 sentences and 3 references. This is doing a disservice to the field of research that currently exists on this topic. I would like to see the introduction greatly expanded upon with regards to EMS training and mechanisms (consider including the work of Hortobagyi & Maffiuletti, 2011, EJAP). Additionally, the authors don’t give any reason as to why EMS was applied to the foot sole (rather than the lower or upper legs, for example). This would be a good opportunity to provide rationale for that decision.

Response: Thank you for your suggestions. We have added this information, as follows.

L. 55–61.

Electrical muscle stimulation (EMS) is an attractive alternative to achieve muscle contraction, thereby acting as a surrogate for habitual physical activity during periods of muscle disuse due to illness or injury [6]. Hortobágy and Maffiuletti et al. reported that EMS modifies the excitability of specific neural pathways and increases maximal voluntary force without apparent muscle hypertrophy [7]. Research has shown that EMS may improve muscle mass [8] and reverse muscle atrophy caused by inactivity without voluntary effort [9].

L. 67–77.

We have previously reported that EMS interventions on the thigh in elderly people did not improve balance performance (single leg stance test) [10]. In addition to the thigh muscles, the plantar intrinsic muscles are also important for balance function [11,12]. These findings indicate the importance of interventions on the plantar intrinsic muscles to promote balance function. Therefore, we focused on EMS of the foot sole to promote muscle contraction in the entire lower limb, including the thigh and lower leg. An EMS device (SIXPAD Foot Fit, MTG Co., Ltd., Nagoya, Japan) was used to apply EMS to the foot sole to stimulate the contraction of muscles in the entire lower limb, mainly the planter intrinsic muscles, tibialis anterior muscle, and triceps surae muscle. We hypothesized that this EMS intervention would be more effective for improving performance in the balance task than thigh stimulation.

The conclusion is lacking in depth. The fact that the control group but not the intervention group demonstrated significant correlations should be investigated further. The conclusion would also benefit from background research on the mechanisms of EMS training and why the hypotheses were made; as well as information on how other training forms have impacted muscle mass, BBS, and FIM in older adults.

Response: Thank you for your suggestion. We have added this information, as follows.

L. 254–261.

Our results support our hypothesis that the EMS intervention on the plantar intrinsic muscles would produce favorable results in terms of balance function. Furthermore, this study demonstrates that EMS interventions can be performed in not only healthy individuals but also those with dementia without adverse events. These findings suggest that EMS is a useful technique for maintaining and improving function in older adults with dementia. Further studies need to examine the area over which EMS is applied, the frequency of application, and the duration of EMS for effective intervention.

Specific:

Experimental Design (line 95): please clarify here if participants were already participating in an exercise intervention (such as the transfer, balance, and gait training) prior to being enrolled in the study, or if all interventions were novel upon recruitment.  It may be helpful to describe participants baseline ‘exercise’ status prior to recruitment, if possible (e.g., participation in nursing home rehabilitation or activities).

Response: Thank you for your suggestion. The participants underwent rehabilitation at the nursing home. This description can be found in section 2.2. Participants, L. 92–93.

Experimental Design (line 96): please clarify here if the EMS and general rehab training were performed in tandem, on the same days but with separation, on different days, etc.

Response: Thank you for your suggestion. We have added this information, as follows.

L. 116–117.

The EMS intervention and general rehabilitation were performed on the same days.

Experimental Design (line 103): please clarify if the measurements (baseline and 12-weeks) were conducted on days when participants had completed training (EMS or general rehab) or not.

Response: Thank you for your suggestion. We have added this information, as follows.

L. 122–124.

Baseline assessments were conducted one week prior to the start of the intervention, and postintervention assessments were conducted within one week after the end of the intervention.

Muscle mass (line 107): please clarify the segmental muscle mass that was calculated (i.e., right and left legs from hip to toe; from knee to toe; whole body?).

Response: Thank you for your suggestion. Muscle mass was calculated for the bilateral thighs and lower legs. We have added this information, as follows.

L. 126–128.

Measurements of lower limb (bilateral thighs and lower legs) muscle mass were performed using direct segmental multifrequency bioelectrical impedance analysis (InBody S10, InBody Japan, Tokyo, Japan).

Statistics: Please clarify if the MMSE score (taken a baseline and 12-weeks) was examined using an ANOVA; and if not, why?

Response: Thank you for your suggestion. The MMSE scores were not analyzed because there were no subjects with changes in this parameter, as described on L. 167.

Figure 2: the small figures, scale, and small adaptations pre-post make it very difficult to see any changes present between groups. Please consider changing the types of graphs (e.g., line graphs from pre-post with a solid vs dotted line for control vs intervention, or bar graph with control vs intervention on the x-axis and the pre-post bars side-by-side for better comparison).

Response: Thank you for your suggestion. We have changed Figure 2 and added information, as follows.

Figure 2. Comparison of muscle mass (A), the BBS score (B), and the FIM score (C) pre- and postintervention between the two groups. * p < 0.05. Data are shown as the mean ± standard error of the mean.

Figures 3 & 4 and respective text: it is unclear why the correlation between BBS and MMSE in the intervention group (fig 3c) and BBS and FIM in the intervention group (fig 4a) are not included in the figures.

Response: Thank you for your suggestion. Since none of the subjects in the intervention group showed a decrease in the BBS score, the ΔBBS score of all the subjects was zero, and an analysis of the correlation coefficient could not be performed; thus, it was not appropriate to include data from the intervention group in Figure 3C or 4A. We have added this information, as follows.

L. 194–195.

Data from the intervention group are not shown in Figure C because none of the individuals in this group had a decreased BBS score.

L. 198–199.

Data from the intervention group are not shown in Figure A because none of the individuals in this group had a decreased BBS score.

Line 184: this is very oddly worded “… and did not decrease balance function in older adults…”. This implies that your hypothesis was that EMS training would decrease balance function. It was not a function of the control intervention (i.e., general rehab) that decreased balance function, but likely a function of the added EMS training which prevented balance decline. This should be reworded to not that EMS training may prevent balance decline or maintain current balance function.

Response: Thank you for your suggestion. We have revised the text, as follows.

L. 203–206.

The primary results of the present study were that a combined EMS intervention on the lower limbs increased muscle mass and prevented balance decline or maintained current balance function in adults aged over 80 years with dementia.

Line 215: This sentence requires greater clarity – what do you mean by ‘did not show improved physical function’? Does this refer specifically to the FIM measure. And to further clarify, please specify by intervention group that you mean the EMS training group.

Response: Thank you for your suggestion. This refers to the FIM score. We have revised the text, as follows.

L. 236–237.

However, our findings did not show improved FIM scores (ADL levels) in either group.

Reviewer 2 Report

I have reviewed the following manuscript “The effect of electrical muscle stimulation on muscle mass and balance in older adults with dementia”

The idea and concept of the study is well-thought and the approach of electrical stimulation

However, the data shown in the graphs and the results, which are demonstrated in the result section do not align.

I have a major concern about the data analysis or interpretation part especially for the Figure 2.  By looking at the graphs (in figure 2), the difference between the pre and post intervention does not seem to be significant. Bar graphs in the Figure 2A (muscle mass), 2B( BBB), and 2C ( FIM) are almost same hence I am afraid whether there is any statistical significance. However, author claims otherwise.

Study claims the a strong correlation between MMSE and BBB though controls are missing from the data, Figure 3C.

Study claims the a strong correlation between FIM and BBB though controls are missing from the data, Figure 4A.

Based on the data presented, I would be afraid to draw an conclusion that EMS intervention increased muscle mass and promoted balance performance in the older adults with dementia.  

Moderate english proofreading would add up clarity in understanding the interpretation of the data.

Author Response

Response to reviewer

We thank the reviewers for their thorough review of our manuscript. We have substantially revised our manuscript according to the suggestions of reviewer. Point-by-point responses to each of the reviewers’ comments are provided below.

Reviewer 2

I have reviewed the following manuscript “The effect of electrical muscle stimulation on muscle mass and balance in older adults with dementia”

The idea and concept of the study is well-thought and the approach of electrical stimulation

However, the data shown in the graphs and the results, which are demonstrated in the result section do not align.

I have a major concern about the data analysis or interpretation part especially for the Figure 2.  By looking at the graphs (in figure 2), the difference between the pre and post intervention does not seem to be significant. Bar graphs in the Figure 2A (muscle mass), 2B(BBB), and 2C (FIM) are almost same hence I am afraid whether there is any statistical significance. However, author claims otherwise.

Response: Thank you for your suggestion. As noted, the change after compared with before the intervention was not large. However, almost all participants in the intervention group showed increased muscle mass (95% CI = 0.18 to 0.69, effect size = 0.51), and almost all participants in the control group showed decreased BBS scores (95% CI = -1.26 to -0.02, effect size = 0.59). We consider these results meaningful. We have revised Figure 2 according to the suggestions of another reviewer.

Study claims the a strong correlation between MMSE and BBB though controls are missing from the data, Figure 3C.

Study claims the a strong correlation between FIM and BBB though controls are missing from the data, Figure 4A.

Response: Thank you for your suggestions. Since none of the subjects in the intervention group showed a decrease in the BBS score, the ΔBBS of all the subjects was zero, and an analysis of the correlation coefficient could not be performed; thus, it was not appropriate to include data from the intervention group in Figure 3C and 4A. We have added this information, as follows.

L. 194–195.

Data from the intervention group are not shown in Figure C because none of the individuals in this group had a decreased BBS score.

L. 198–199.

Data from the intervention group are not shown in Figure A because none of the individuals in this group had a decreased BBS score.

Based on the data presented, I would be afraid to draw an conclusion that EMS intervention increased muscle mass and promoted balance performance in the older adults with dementia.

Response: Thank you for your suggestion. It has already been reported that EMS interventions can contribute to muscle mass gain in immobile patients (Dirks et al. 2014) and healthy older adults (aged 75 to 96 years old) (Benavent-Caballer et al. 2014, Nishikawa et al. 2019), and the results of this study are similar to those results. Although our study’s subjects were slightly older (aged 80 to 100 years old) than the subjects in these previous studies, our results suggest that EMS interventions are effective for improving physical function in older adults. We consider that our results indicate the possibility of a new intervention approach in older adults with dementia.

Moderate english proofreading would add up clarity in understanding the interpretation of the data.

Response: Thank you for your suggestion. The manuscript was edited again by American Journal Experts.

Round 2

Reviewer 2 Report

I had a chance to review the revised version of your manuscript. Based on my previous comments, I see a reasonable amount of changes and justification for this work.

This manuscript is a resubmission of an earlier submission. The following is a list of the peer review reports and author responses from that submission.